# Simultaneous Visualization of Lung Tumor and Intersegmental Plane during Pulmonary Segmentectomy by Intravenous Injection of Indocyanine Green

**DOI:** 10.3390/cancers16071439

**Published:** 2024-04-08

**Authors:** Kyungsu Kim, Ok Hwa Jeon, Byeong Hyeon Choi, Jiyun Rho, Jun Hee Lee, Jae Seon Eo, Beop-Min Kim, Hyun Koo Kim

**Affiliations:** 1Department of Thoracic and Cardiovascular Surgery, Korea University Guro Hospital, Korea University College of Medicine, Seoul 08308, Republic of Korea; k_gangter@korea.ac.kr (K.K.); hwa1983418@korea.ac.kr (O.H.J.); bhchoi1001@korea.ac.kr (B.H.C.); jiyunr1219@korea.ac.kr (J.R.); jhl3728@korea.ac.kr (J.H.L.); 2Department of Biomedical Sciences, Korea University College of Medicine, Seoul 02841, Republic of Korea; 3Department of Nuclear Medicine, Korea University Guro Hospital, Korea University College of Medicine, Seoul 08308, Republic of Korea; jaeseon76@gmail.com; 4Department of Biomedical Engineering, Korea University College of Health Science, Seoul 02841, Republic of Korea; bmk515@korea.ac.kr; 5Interdisciplinary Program in Precision Public Health, Korea University, Seoul 02841, Republic of Korea

**Keywords:** indocyanine green, fluorescence imaging, segmentectomy, lung cancer

## Abstract

**Simple Summary:**

The key to successful segmentectomy lies in precisely identifying the tumor and intersegmental planes to ensure adequate resection margins. This study is the first to use intravenous indocyanine green injections at different time points and doses to simultaneously detect lung cancer and intersegmental planes. Additionally, we aimed to assess the clinical feasibility of the method through a combination of preclinical and clinical studies. This approach not only simplifies the treatment process for patients by avoiding complex procedures but also accurately defines the resection margins for lung cancer while addressing the complications arising from preoperative CT-guided positioning. This technique has the potential to improve patient outcomes and improve the quality of life for patients with lung cancer.

**Abstract:**

Segmentectomy is a targeted surgical approach tailored for patients with compromised health and early-stage lung cancer. The key to successful segmentectomy lies in precisely identifying the tumor and intersegmental planes to ensure adequate resection margins. In this study, we aimed to enhance this process by simultaneously visualizing the tumor and intersegmental planes through the intravenous injection of indocyanine green (ICG) at different time points and doses. Lung tumors were detected by intravenous injection of ICG at a dose of 2 mg/kg 12 h before surgery in a rabbit model. Following the dissection of the pulmonary artery, vein, and bronchi of the target segment, 0.6 mg/kg of ICG was injected intravenously to detect the intersegmental plan. Fluorescent images of the lung tumors and segments were acquired, and the fluorescent signal was quantified using the signal-to-background ratio (SBR). Finally, a pilot study of this method was conducted in three patients with lung cancer. In a preclinical study, the SBR of the tumor (4.4 ± 0.1) and nontargeted segments (10.5 ± 0.8) were significantly higher than that of the targeted segment (1.6 ± 0.2) (targeted segment vs. nontarget segment, *p* < 0.0001; target segment vs. tumor, *p* < 0.01). Consistent with preclinical results, lung tumors and the intersegmental plane were successfully detected in patients with lung cancer. Consequently, adequate resection margins were identified during the surgery, and segmentectomy was successfully performed in patients with lung cancer. This study is the first to use intravenous ICG injections at different time points and doses to simultaneously detect lung cancer and intersegmental planes, thereby achieving segmentectomy for lung cancer.

## 1. Introduction

Lung cancer is the leading cause of cancer-related deaths worldwide [1,2]. Recent advancements in computed tomography (CT) screening have markedly increased the detection of lung cancers with small lung nodules, offering a promising avenue to significantly reduce mortality rates [3]. Surgery is the preferred treatment for ensuring the long-term survival of patients with early-stage nonsmall cell lung cancer (NSCLC) [4,5]. Clinical trials, specifically focusing on the limited resection of lung cancer, revealed that the 5-year overall survival rate of lung cancer patients with a tumor diameter of ≤2 cm who underwent sublobar resection is 1.4–3.2% higher than that of patients who underwent lobectomy [6,7]. For patients diagnosed with extremely small early-stage lung cancer [8,9,10,11], limited resection during cancer surgery, such as segmentectomy and wedge resection, has demonstrated comparable overall survival to lobectomy [6,7]. In addition, considering that the median age at lung cancer diagnosis is 70 years, which is usually accompanied by numerous comorbidities, limited cardiac function, and poor lung function [12], a limited and curative resection, such as segmentectomy, could be performed as an alternative for these compromised patients.

Anatomical segmentectomy is considered one of the most effective surgical methods for lung cancer resection. This method enables segmental lymph node dissection, provides safer surgical margins, and lowers local recurrence rates [8]. However, a critical aspect of successfully performing segmentectomy is the accurate identification of the intersegmental plane where the tumor is located. During surgery, particularly in robotic or video thoracoscopic surgery, the challenge lies not only in identifying the lung tumor but also in accurately locating the intersegmental plane.

Indocyanine green (ICG) can be used as a fluorescent dye and is the infrared fluorescence contrast agent approved by the Food and Drug Administration. ICG was first used for hepatic function diagnostics and has been introduced as a useful tool for the real-time visualization of lymphatic flow and tissue perfusion during surgery [13]. Additionally, ICG was first utilized for intraoperative observation of hepatocellular carcinoma and has since been employed for the detection of various types of cancer, including pulmonary nodules [14,15,16]. Recently, the intravenous injections of ICG within 12–24 h before surgery for the accurate detection of tumor margins have been investigated [2,15,16,17]. In our previous study, we used intravenous ICG within 12 h before surgery to detect and successfully remove lung tumors. Therefore, intravenous ICG-based near-infrared (NIR) fluorescence imaging is an applicable tool for the intraoperative identification of lung tumors.

In segmentectomy, invasive preoperative localization methods such as hook-wire, micro coil, and combinations of materials such as fluorescent iodinate emulsions at the tumor site are widely used [15,18,19]. However, the use of ICG-lipiodol emulsion for pulmonary nodule localization only provides an image of the material rather than the tumor itself, making it challenging to accurately identify the resection margin [20]. Moreover, CT-guided positioning prolongs patient waiting times, thereby exacerbating patient anxiety and causing related complications such as pneumothorax and pulmonary hemorrhage [20,21,22]. 

Traditionally, the identification of the intersegmental plane involved determining the segmental planes by inflating and deflating the targeted segment through clamping and unclamping the relevant bronchus [23]. However, an inflated lung may obstruct the view of the target segment during video-assisted thoracoscopic surgery (VATS) or robot-assisted thoracoscopic surgery (RATS). Many studies have reported the feasibility of using intraoperative intravenous ICG to detect intersegmental planes, facilitating the swift and straightforward identification of the targeted lung segment without requiring lung inflation [24,25,26,27]. In our previous study, this method of intersegmental plane identification was used in combination with lung tumor localization by ICG-lipiodol emulsion to visualize the lung tumor and intersegmental plane during segmentectomy [28]. However, to reduce patient inconvenience, mitigate complications, and alleviate the burden on medical staff, further enhancement of this method is necessary.

Despite the high applicability of intravenous ICG at different time points in the detection of lung tumor margins and intersegmental planes during lung cancer surgery owing to its convenience, no studies have combined these two methods for lung cancer surgery. In this study, to improve the convenience and safety of patients and healthcare professionals, we developed methods to simultaneously identify lung tumors and intersegmental planes. This was achieved through the use of preoperative and intraoperative intravenous ICG. Subsequently, we evaluated its applicability for lung cancer surgery by conducting preclinical and pilot trials. We hope that the simultaneous visualization of lung tumors and intersegmental planes will enable complete tumor resection with adequate negative margins, thereby improving the quality of life and survival rate of patients with cancer.

## 2. Materials and Methods

### 2.1. Preparation of ICG

ICG (25 mg vials; JEIL, Seoul, Republic of Korea) was dissolved in 10 mL of injectable stock solution (2.5 mg/mL). In a previous study, we showed that 2 mg/kg of ICG injected 12 h before surgery was the optimal injection method for tumor detection [29], and a 0.3–0.6 mg/kg dose was used for identifying intersegmental planes [30]. Therefore, 2 mg/kg of ICG was intravenously injected 12 h prior to surgery for the detection of lung tumors, while 0.3–0.6 mg/kg of ICG was intravenously administered during surgery for identifying intersegmental planes. 

### 2.2. Rabbit Model with VX2 Lung Cancer

All animal experiments and protocols were approved by the Animal Experimentation Committee of the Korea University Medical Center and carried out in accordance with the institutional guidelines for the care and use of laboratory animals. Twelve female New Zealand white rabbits (aged 8 weeks, 2.5–3 kg, DooYeol Biotech Co., Ltd., Seoul, Republic of Korea) were used in this study. The rabbits were maintained under controlled temperature, humidity, and illumination.

Twelve rabbit models of VX2 lung cancer were evaluated in a previous study [31]. For tumor transplantation, fresh VX2 tissue was harvested using a surgical blade and washed with phosphate-buffered saline (PBS), and any surrounding necrotic tumor tissue was removed. The tissues were then sliced into small pieces and filtered using a 100-μm cell strainer to obtain a VX2 suspension. The suspension was centrifuged at 1200 rpm for 3 min and resuspended in PBS at a concentration of 1 × 10^7^ cells/mL. The VX2 carcinoma was mixed with 100 μL Matrigel and prepared in a 1-mL syringe with a 23-gauge needle. All rabbits were anesthetized with intramuscular injections of xylazine (5 mg/kg; Rompun^TM^, Bayer Korea Inc., Seoul, Republic of Korea) and tiletamine-zolazepam (10 mg/kg; Zoletil 50; Virbac Korea Inc., Seoul, Republic of Korea). Chest hair was shaved and sterilized with povidone-iodine for disinfection. A mixture of VX2 carcinoma and Matrigel was injected into the rabbits. A rabbit model of VX2 lung cancer was established 2–3 weeks postinjection and confirmed using positron emission tomography-computed tomography (PET/CT).

### 2.3. Lung Tumor and Pulmonary Segmental Margin Detection in Rabbits Using ICG

The established rabbit model of VX2 lung cancer was anesthetized using xylazine and tiletamine-zolazepam, and ICG (2 mg/kg) was administered into the ear vein. Open thoracotomy was performed 12 h after ICG injection and lung tumors were identified using an intraoperative color and fluorescence-merged imaging system (ICFIS), as previously described [32]. The targeted segmental artery, vein, and bronchus were ligated, and an additional 0.6 mg/kg was injected intravenously. The real-time visualization of the intersegmental plane was accomplished using ICFIS, and the targeted segment was surgically removed. The resected lung tumor tissues were stained with hematoxylin and eosin (Dako, Glostrup, Denmark) for histological analysis. The signal-to-background ratio (SBR) in tumors, nontargeted segments, and targeted segments was quantified using the ImageJ software (64-bit Java 1.8.0_172, National Institute of Healthcare, Bethesda, MD, USA).

### 2.4. Patients

This study was approved by the Institutional Review Board of the Korea University Guro Hospital (IRB No. 2020GR0045). All patients were diagnosed with lung cancer and scheduled for segmentectomy at Korea University Guro Hospital. In our hospital, segmentectomy was indicated for patients with a solitary peripheral pulmonary nodule with a solid part of <2 cm, a consolidation-to-tumor ratio > 0.5, GGN, and no lymph node metastasis on preoperative CT. In addition, patients with centrally located metastatic lung cancer or poor pulmonary reserve required segmentectomy instead of lobectomy to preserve the lung parenchyma. Patients with liver dysfunction (aspartate transaminase and alanine transaminase levels of >2.5 times the normal value), hypersensitivity, or adverse reactions to ICG and those receiving neoadjuvant chemotherapy were excluded.

### 2.5. Surgical Procedures

All patients underwent pulmonary segmentectomy with VATS or RATS, a major procedure for treating lung cancer [29,33]. A lung tumor was confirmed through firefly fluorescent imaging using Pinpoint^®^ thoracoscope (Stryker Corp., Kalamazoo, MI, USA) or the da Vinci system (Intuitive Surgical, Inc., Sunnyvale, CA, USA) immediately after surgery. Following the ligation of the pulmonary segmental artery, vein, and bronchus of the targeted segment, 0.35 mg/kg of ICG was injected into the systemic vein. Subsequently, the target segment containing the tumor was identified by detecting variations in the NIR fluorescence signals compared with the nontarget segments. Therefore, surgeons utilized NIR fluorescence images to identify lung tumors and intersegmental planes and used a narrow endostapler (ECHELON FLEX™ Powered Vascular Stapler, Ethicon Inc., New Bridgewater, NJ, USA) or a robotic stapler (EndoWrist Stapler; Intuitive Surgical Inc., Sunnyvale, CA, USA) to divide the intersegmental planes, ensuring a distance of >2 cm between the tumor and the resection line or a resection larger than the tumor, thereby completing pulmonary segmentectomy. Routine systemic lymph node dissection was performed during segmentectomy.

### 2.6. Statistical Analysis

One-way analysis of variance was used to analyze the differences in SBR between the tumor, targeted, and nontargeted segments in the rabbit tumor model and patients with lung cancer. All graphs, calculations, and statistical analyses were performed using the GraphPad Prism software version 8.4.3 (Windows GraphPad Software, San Diego, CA, USA). A *p*-value of <0.05 was considered significant.

## 3. Results

### 3.1. Simultaneous Identification of Intersegmental Plane and Tumor in a Rabbit Tumor Model

In all three rabbits, lung cancer models were effectively established and validated using PET/CT imaging and histological examination (Figure 1A,B). The mean tumor size was 1.0 ± 0.2 cm (range, 0.8–1.2 cm). Lung tumors were successfully detected in all rabbits using fluorescence imaging (Figure 2A). Simultaneously, the intravenous injection of ICG during surgery created distinguishable differences in the NIR fluorescence signals between the target and nontarget segments, facilitating the clear identification of the intersegmental plane (Figure 2A). Consistent with the fluorescence image, the SBR of the tumor (4.4 ± 0.1) and nontargeted segments (10.5 ± 0.8) was significantly higher than that of the targeted segment (1.6 ± 0.2) (targeted segment vs. nontarget segment, **** *p* < 0.0001; target segment vs. tumor, ** *p* < 0.01) (Figure 2B). Therefore, the intravenous injection of ICG at different time points enables the simultaneous observation of the intersegmental plane and tumor.

### 3.2. Characteristics of Patients

Three patients with lung cancer, with a mean age of 68 (range: 57–75 years) were enrolled in the pilot trial. Pulmonary nodules measuring 2.3 ± 0.2 cm (range: 2.1–2.5 cm) in size were detected on chest CT. Despite all patients having tumors of >2.0 cm, one had poor pulmonary reserve, while the other presented with metastatic lung cancer from the central portion of the endometrium and the liver, respectively. Consequently, segmentectomy was performed in all patients. The pulmonary nodules were identified in the left lower lobe and left upper lobe. Following segmentectomy, thorough pathological analyses revealed the presence of lung adenocarcinoma with a tumor size of 2.3 cm in one patient, with a pathological TNM stage of T2aN0M0. The other patient was diagnosed with metastatic endometrial stromal sarcoma with a tumor size of 3.0 cm and hepatocellular carcinoma with a tumor size of 2.2 cm, respectively. All the resection margins exceeded the established threshold of 2.0 cm. No intra- or postoperative complications related to ICG administration were observed. The patient was discharged without any complications. The lengths of hospital stay were 4–5 d, and both patients did not require additional treatment.

### 3.3. Simultaneous Visualization of the Intersegmental Planes and Tumor in Patients

Employing the optimal technique for the intravenous injection of ICG, as established in preclinical studies, we assessed its feasibility for simultaneously visualizing lung tumors and intersegmental planes in all patients with lung cancer. Consistent with the results of preclinical studies, the lung tumor and intersegmental planes were successfully determined during surgery in a patient with lung cancer, and segmentectomy was performed (Figure 3A). The SBR of the tumor (3.9 ± 1.2) and nontargeted segments (4.0 ± 0.8) were significantly higher than that of the targeted segment (1.3 ± 0.2) (targeted segment vs. nontarget segment, ** *p* < 0.01; targeted segment vs. tumor, ** *p* < 0.01) (Figure 3B).

## 4. Discussion

Our study developed a noninvasive method using intravenous ICG injections, enabling the simultaneous identification of lung tumors and intersegmental planes. This approach eliminates the need for preoperative CT-guided pulmonary nodule localization, thereby enhancing patient convenience, reducing complications, and alleviating the burden on medical staff. 

Given the increasing proportion of NSCLC patients with small early-stage lung tumors or those with limited functional capacity, the adoption of segmentectomy has the potential to improve patients’ postoperative quality of life [8,9,10,11]. Successful segmentectomy execution necessitates the simultaneous identification of the lung tumor and intersegmental plane during surgery. Many researchers have recognized the importance of identifying lung tumors and intersegmental planes [26,30,31,32,34,35]. Most studies have only identified lung tumors or intersegmental planes. No existing studies have utilized intravenous ICG to concurrently detect both of these elements [26,32,34,35]. In our previous study, we simultaneously identified lung tumors and intersegmental planes using the local injection of ICG and an iodized emulsion via preoperative CT and intraoperative intravenous ICG, respectively. However, the preoperative CT-guided local injection of ICG and iodinated emulsion solely provides an image of the material rather than the tumor itself. Moreover, this approach poses potential complications such as pneumothorax, intraparenchymal hematoma, pleural reaction, and air leak [36,37,38]. 

In this study, we developed a simple, noninvasive method of intravenous injection of ICG at different time points to simultaneously identify lung tumors and intersegmental planes. In our preclinical study, we detected lung tumors and intersegmental planes in a rabbit model. Moreover, we successfully identified both lung tumors and intersegmental planes in patients with lung cancer and performed a segmentectomy. The intravenous injection of ICG instead of CT-guided positioning as a tumor-marking method eliminates the complicated process of marking tumors. 

Considering the limitations of detecting deep NIR fluorescence signals beyond a certain tissue depth, we established rabbit models of lung tumors to evaluate the feasibility of our method for ICG image-guided lung cancer surgery. Lung tumors were identified by intravenous injection of 2 mg/kg of ICG 12 h before surgery. To identify intersegmental planes, a 0.6 mg/kg dose of ICG was injected, determined as the optimized dose in previous studies using the ICFIS system [39]. Consequently, the lung tumors and intersegmental planes in rabbit models of superficial lung tumors were simultaneously visualized.

More importantly, we demonstrate the feasibility of this method in three patients with lung cancer. Lung tumors were identified through the intravenous injection of 2 mg/kg of ICG 12 h before surgery. To identify intersegmental planes, a 0.35 mg/kg dose of ICG was injected, previously confirmed as the optimal dose utilizing the da Vinci system [28,33]. Intravenous ICG dosage variations during surgery between animal experiments and clinical trials are likely attributable to device performance and species differences. The intravenous administration of different ICG doses at different time points facilitated the accurate visualization of both lung tumors and intersegmental planes, leading to successful segmentectomy. Despite the limited number of patients included in this study, the detection of lung tumors and intersegmental planes in both patients suggests the promising potential for the successful application of this method in patients. 

Although different time-point ICG injections prove effective in facilitating segmentectomies, they have several limitations. First, the applicability is constrained to be located on or near the surface of the lung/fissure tumors owing to the depth issue of ICG. Therefore, for deep-seated tumors, we apply preoperative CT-guided localization following mapping with a 3D image analysis system for surgical planning. Second, as this study was a pilot trial with a limited number of patients, assessing the feasibility of this approach using a larger sample size is crucial. Finally, because ICG is not a lung cancer-specific contrast agent, further studies using cancer-specific targeted contrast agents such as OTL38 are necessary. 

## 5. Conclusions

This study is the first to develop and evaluate a noninvasive method of intravenously injecting an ICG at different time points to identify both lung tumors and intersegmental planes. This approach not only simplifies the treatment process for patients by avoiding complex procedures but also accurately defines the resection margins for lung cancer while addressing the complications arising from preoperative CT-guided positioning. This new method will contribute to the image-guided segmentectomy of lung tumors and improve the quality of life and survival rate of patients with lung cancer.

## Figures and Tables

**Figure 1 cancers-16-01439-f001:**
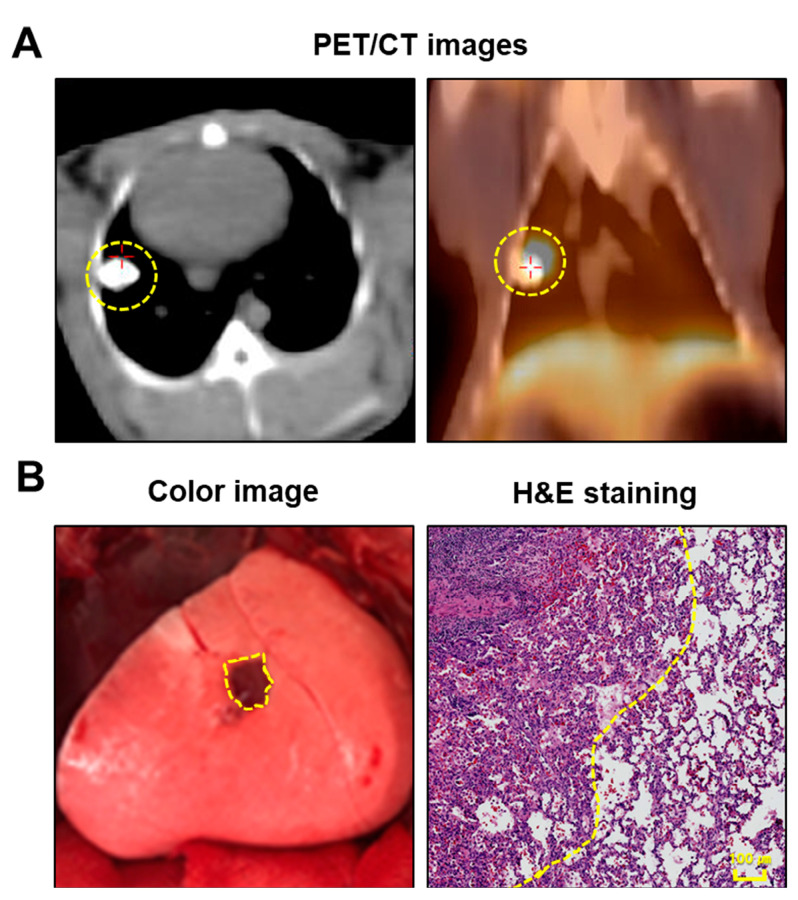
Establishment of the rabbit VX2 lung tumor model. (**A**) Representative PET/CT images of rabbit lung tumor model. VX2 lung tumor can be evaluated at 2–3 weeks postinjection of VX2 tumor. (**B**) Representative intraoperative surgical image of the lung tumor under white light (**left**), and pathology image of the lung tumor with H&E staining. The yellow dotted line indicates a lung tumor. PET/CT, positron emission tomography-computed tomography; H&E, hematoxylin and eosin.

**Figure 2 cancers-16-01439-f002:**
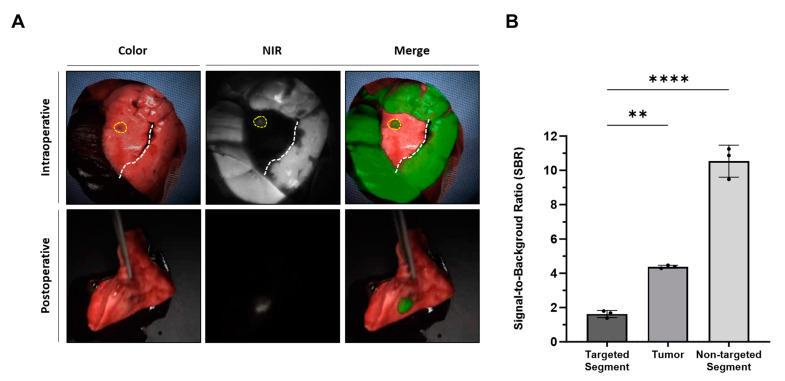
Simultaneous visualization of lung tumor and intersegmental planes with intravenously injected ICG in a rabbit lung tumor model (**A**) Representative intraoperative and postoperative color, NIR, and merged images of a rabbit lung tumor model with intravenous injection of ICG at different time points and doses. The yellow dotted line indicates the lung tumor, while the white dotted line indicates the intersegmental plane. (**B**) SBR of the target segment, tumor, and nontarget segment; statistical analysis was performed using a one-way analysis of variance; targeted segment vs. nontarget segment, **** *p* < 0.0001; target segment vs. tumor, ** *p* < 0.01. NIR, near-infrared; SBR, signal-to-background ratio.

**Figure 3 cancers-16-01439-f003:**
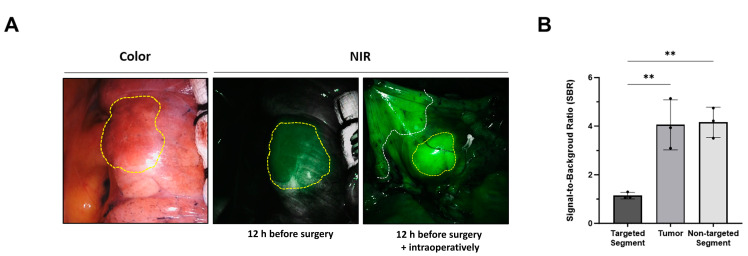
Simultaneous visualization of lung tumor and intersegmental planes with intravenously injected ICG in patients with lung cancer. (**A**) Representative surgical color and fluorescent images from each patient with intravenous injection of ICG at different time points and doses. The yellow dotted line indicates lung tumor, while the white dotted line indicates the intersegmental plane. (**B**) SBR of the targeted segment, tumor, and nontargeted segments. Statistical analysis was performed using one-way analysis of variance; targeted segment vs. nontarget segment, ** *p* < 0.01; targeted segment vs. tumor, ** *p* < 0.01. NIR, near-infrared; SBR, signal-to-background ratio.

## Data Availability

The data presented in this study are available upon request from the corresponding authors.

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
