# Peer review of "Simultaneous Visualization of Lung Tumor and Intersegmental Plane during Pulmonary Segmentectomy by Intravenous Injection of Indocyanine Green"

_cancers, 2024, doi:10.3390/cancers16071439_

Round 1

Reviewer 1 Report

Comments and Suggestions for Authors

The authors deserve credit for the considerable effort to show the effectiveness of ICG injections for localization of the pulmonary tumors and the intersegmental fissures in the experimental model and in 3 human patients.

I was not fully convinced by the authors in their belief that visualization of the tumors was clear, I cannot see that on the photos.

Moreover, the presented technique is able to visualize only the superficially localized tumors which are easy to find intraoperatively and is not very helpful for the deeply localized tumors, which are more difficult.

Reviewer 2 Report

Comments and Suggestions for Authors

Dear Author:

I reviewed the manuscript entitled “Simultaneous Visualization of Lung Tumor and Intersegmental Plane During Pulmonary Segmentectomy by Intravenous Injection of Indocyanine Green.” Thank you for reporting a valuable method. This manuscript was very interesting for a precise segmentectomy because the contents of this manuscript included the problems and novel improvements of the precise segmentectomy, although it has a few limitations.

Comments-

1.      I think that the difficulties of segmentectomy depend on whether the target tumor could be detected during surgery. The difficulties of segmentectomy for the patient whose tumor localization is detected are absolutely different from that the patients whose tumor localization is undetectable. If the tumor can be detected during surgery, the segmentectomy with a sufficient surgical margin would be easy because the intersegmental divisions could be performed while securing a sufficient surgical margin while confirming the tumor location. On the other hand, we sometimes experience cases in which we could not clearly identify the intersegmental line deep in the parenchyma when we use ICG intravenously. If the tumor is located deep in the parenchyma, the tumor would tend to be undetectable during surgery. In such a case, I think that the intersegmental veins would be usable as the demarcation of the intersegmental line. Therefore, while I always perform the segmentectomy in such undetectable tumor cases, referring to the inflation-deflation line and intersegmental veins, I always use the ICG in the cases whose tumor is detectable. In this report, it was unclear if the target tumor could be detected without the ICG method, if this method could identify the target nodules located deep in the parenchyma, and if this method was applied in all cases or selective cases depending on the tumor localization. I hope that the authors describe these points.

2.      In Figure 3, the distance from the yellow dots to the white dots looks like to be very narrow. Is the surgical margin sufficient in this case? 

Yours sincerely,

Reviewer 3 Report

Comments and Suggestions for Authors

The authors describe a preclinical rabbit model of simultaneous visualisation of lung tumour and the intersegmental planes using IV injection of indocyanine green. This is to assist in doing a complete segmentomy. They subsequently used this approach in surgical resection of tumour from lung in 3 patients. This is an interesting approach and deserves to be further explored.

Questions

1) It was mentioned towards the end of the paper that this technique is only relevant to tumours on the surface or near the surface of the lung / fissure. What proportion of patients would this be applicable to?

2) The clinical sample size of 3 lung tumours, 2 of which weren't pulmonary primary neoplasms, provides very preliminary information. In the first instance, applying this technique to varying sizes of primary lung neoplasms in patients is needed. In addition, a range of histological subtypes from lepidic to acinar to solid may well vary in signal to background ration.

3) There may also be differences in the success of this technique across a range of metastases from extra-thoracic neoplasms. For example sarcoma including osteosarcoma, adenocarcinoma from bowel etc.

4) Are there potential side effects of this dye?   

5) The final sentence of the Simple Summary - this is an overstatement. It would be reasonable to state that ' this technique has the potential to improve patient outcomes"
